# Evaluation of Firefly and *Renilla* Luciferase Inhibition in Reporter-Gene Assays: A Case of Isoflavonoids

**DOI:** 10.3390/ijms22136927

**Published:** 2021-06-28

**Authors:** Maša Kenda, Jan Vegelj, Barbara Herlah, Andrej Perdih, Přemysl Mladěnka, Marija Sollner Dolenc

**Affiliations:** 1Faculty of Pharmacy, University of Ljubljana, Aškerčeva Cesta 7, 1000 Ljubljana, Slovenia; masa.kenda@ffa.uni-lj.si (M.K.); jan.vegelj@gmail.com (J.V.); barbara.herlah@ki.si (B.H.); andrej.perdih@ki.si (A.P.); 2National Institute of Chemistry, Hajdrihova 19, 1000 Ljubljana, Slovenia; 3Faculty of Pharmacy in Hradec Králové, Charles University, Akademika Heyrovského 1203, 500 05 Hradec Králové, Czech Republic; mladenkap@faf.cuni.cz

**Keywords:** isoflavonoids, firefly luciferase, *Renilla* luciferase, inhibition, quantitative structure–activity relationships, molecular docking, reporter-gene assay, OECD test guidelines, alternative methods

## Abstract

Firefly luciferase is susceptible to inhibition and stabilization by compounds under investigation for biological activity and toxicity. This can lead to false-positive results in in vitro cell-based assays. However, firefly luciferase remains one of the most commonly used reporter genes. Here, we evaluated isoflavonoids for inhibition of firefly luciferase. These natural compounds are often studied using luciferase reporter-gene assays. We used a quantitative structure–activity relationship (QSAR) model to compare the results of in silico predictions with a newly developed in vitro assay that enables concomitant detection of inhibition of firefly and *Renilla* luciferases. The QSAR model predicted a moderate to high likelihood of firefly luciferase inhibition for all of the 11 isoflavonoids investigated, and the in vitro assays confirmed this for seven of them: daidzein, genistein, glycitein, prunetin, biochanin A, calycosin, and formononetin. In contrast, none of the 11 isoflavonoids inhibited *Renilla* luciferase. Molecular docking calculations indicated that isoflavonoids interact favorably with the D-luciferin binding pocket of firefly luciferase. These data demonstrate the importance of reporter-enzyme inhibition when studying the effects of such compounds and suggest that this in vitro assay can be used to exclude false-positives due to firefly or *Renilla* luciferase inhibition, and to thus define the most appropriate reporter gene.

## 1. Introduction

Advances in high-throughput screening of compounds for biological activity have enabled the compilation of data from cell-based and biochemical assays on a myriad of biological targets for thousands of compounds, such as the US Federal Consortium on Toxicology in the 21st Century (Tox21 Consortium) program [1,2]. In this way, our knowledge of the biological activities of compounds has expanded rapidly, whereby the applicability of a compound (e.g., as a drug) and its potential toxicity can be identified early in the process of drug discovery. Biological activities are most often detected in assays using fluorescence (as ~53% of cases), followed by luminescence (~21%), and absorbance (<7%) [3]. The interpretation of the data obtained can often, however, be challenging, due to interference in the assays, such as quenching and autofluorescence in fluorescence-based assays [1], inhibition and stabilization of the enzyme in luminescence-based assays [3,4,5], and interference of colored compounds in absorbance-based assays [6]. All such situations can lead to false-positive results. An additional problem in cell-based fluorescence assays is the possible interference from cell autofluorescence [7], although this is not a problem in cell-based bioluminescence assays.

Luciferase reporter-gene assays are a commonly used bioluminescence assay. The construct includes a promoter region of a gene of interest followed by a luciferase gene. When this is introduced into a cell, luciferase is expressed in quantities that are proportional to the promoter activity. The luciferase (and hence the promoter activity) can then be quantified by the measurement of the luminescence produced when the enzyme substrate is added. In this way, the transcriptional activity of the gene of interest (i.e., its expression) can be measured in response to the effects of different modulators of the relevant signaling pathways [8]. The luciferase reaction can also be used in combination with constitutively active promoters, to investigate cytotoxicity or transfection efficiency [9,10]. Similarly for cell viability assays for cellular ATP quantification, where in contrast to reporter-gene assays, the luciferase is not produced in the cell [11].

Luciferase from the firefly *Photinus pyralis* (FLuc) is the most widely used luciferase for bioluminescence assays. FLuc requires ATP and oxygen, and it catalyzes the oxidation of D-luciferin to oxyluciferin, with concurrent emission of a photon. However, during testing of 360,000 compounds in a chemical library, FLuc was shown to be inhibited by ~12% [12]. FLuc inhibitors are usually low molecular weight compounds with linear, planar structures that contain benzothiazoles, benzoxazoles, benzimidazoles, oxadiazoles, hydrazines, and/or benzoic acids [12,13]. They compete with the substrate, D-luciferin (e.g., benzothiazoles) and/or with ATP (e.g., hydrazines), through non-competitive mechanisms (e.g., resveratrol) or through the formation of multisubstrate adduct inhibitors [12,14,15]. Paradoxically, FLuc inhibitors can also increase the luminescence produced, due to the stabilization of the enzyme structure. Furthermore, a stable complex between an inhibitor and FLuc can be formed, thereby preventing FLuc degradation, which leads to its accumulation in cells [3].

Luciferase from the sea pansy *Renilla reniformis* (RLuc) is also used in reporter-gene assays, and frequently in combination with FLuc, to determine cytotoxicity while providing for normalization of the results. Such dual-luciferase assays are possible due to the different substrate requirements of FLuc and RLuc, and to the possibility to quench FLuc luminescence before adding the RLuc substrate, coelenterazine [3]. RLuc is ATP-independent and has been shown to be less susceptible to inhibition by known FLuc inhibitors [13]. However, there are also many inhibitors of RLuc, as an estimated 10% of chemical libraries; however, target interference in RLuc assays has been far less studied than for FLuc assays [3].

The effects of luciferase inhibition are not only seen in terms of the isolated enzyme or when it is in cell lysates, but also in assays with viable cells, where any inhibitor first needs to pass through the cell membrane, to enter the cell, as for reporter-gene assays [16]. The potential for such interference in luciferase assays is often neglected, and indeed, some known FLuc inhibitors, such as resveratrol, are still used in cell-based assays with FLuc as the reporter gene [14,17]. Moreover, as for resveratrol, other stilbene analogs were shown to inhibit FLuc, which is still used in several assays to screen these compounds for endocrine activity; this is particularly concerning, as stilbene derivatives often have endocrine-disrupting activities [18]. Indeed, FLuc and RLuc are included in the standardized guidelines for the testing of chemicals of the Organisation for Economic Co-operation and Development (OECD), mostly for endocrine toxicity (e.g., TG 455 [19] and TG 458 [10], for transactivation of human estrogen and androgen receptors, respectively) and immunotoxicity (e.g., TG 442D [20], for keratinocyte activation). Therefore, it is particularly important to allow for possible assay interference for the correct interpretation of these results. Indeed, these test guidelines highlight the potential for false-positives due to apparent luciferase inhibition or due to nonspecific increases in luminescence, and they suggest protocols to exclude some false-positive results. However, they do not recommend pre-test screening of such compounds to determine their suitability for these reporter-gene assays. Particular caution can be advised for such testing of phytoestrogen compounds, as these were reported to activate FLuc in a nonreceptor-mediated manner.

Predicting potential inhibition of FLuc (and hence the potential for false-positive results in reporter-gene assays) using different chemoinformatic approaches has been a relatively active field in recent years. This has led to the development of different freely available models that can be used when designing studies using FLuc as the reporter protein, such as Luciferase Advisor [21], ChemFLuc [22], and InterPred [1,23]. We were, however, not able to find any in silico models to predict the potential for inhibition of RLuc.

Here, we have evaluated the potential of selected isoflavonoids (with some as known phytoestrogens) to inhibit FLuc and RLuc activities using in silico and in vitro approaches. As the importance of addressing such reporter-enzyme inhibition has been relatively neglected, our aim was also to highlight this aspect using a well-known group of natural products as examples.

## 2. Results

These isoflavonoids were selected based on their structures, to provide insight into their importance for luciferase inhibition. They were therefore sorted according to five structurally diverse groups, as shown in Figure 1: non-methylated isoflavonoids (daidzein, genistein); O-methylation on the A-ring (glycitein, prunetin); O-methylation on the B-ring (biochanin A, calycosin, formononetin); an isoflavandiol (s-equol); and the glucosides (daidzin, genistin, glycitin). These isoflavonoids were first screened for the prediction of FLuc inhibition in silico, using the Tox21 Consortium QSAR model InterPred [23]. They were then assayed for inhibitory activities against FLuc and RLuc in vitro in a rapid, cost-effective, cell-lysate assay system. Finally, the active isoflavonoids were docked into FLuc and RLuc in silico to determine the mode of binding to the enzyme active sites.

### 2.1. InterPred In Silico Prediction of FLuc Inhibition

The predicted potentials were determined, for the positive control (resveratrol) and these selected isoflavonoids, to interfere with FLuc using the Tox21 Consortium QSAR model InterPred [23], which is based on the results of the in vitro FLuc inhibition assays from the Tox21 Consortium library. InterPred predicted high likelihoods of interference (i.e., red color class) for seven of the 11 isoflavonoids: daidzein, genistein, glycitein, prunetin, biochanin A, calycosin, and formononetin. The remaining four isoflavonoids (i.e., the isoflavandiol and the glucosides) were predicted to have moderate potential to inhibit FLuc (color classes, orange and yellow, respectively). The positive control of the stilbene resveratrol for inhibition of FLuc was also predicted to have a high likelihood of interference (red color class), although this was a little lower than seen for genistein, biochanin A, and formononetin. The full data are given in Table 1.

### 2.2. In Vitro Luciferase Inhibition Assay

The inhibitory activities of these isoflavonoids on the FLuc and RLuc enzymes in vitro were determined in lysates of the AR-EcoScreen cell line. This cell line is stably transfected with the *FLuc* and *RLuc* reporter-genes and is used in OECD TG 458 for compound testing for (anti)androgen activity [10]. Using AR-EcoScreen cell lysates that contained both FLuc and RLuc as the dual luciferase assay reagents allowed their concomitant investigation for the influence of the isoflavonoids. The results for these in vitro FLuc and RLuc inhibition assays for the isoflavonoids at 1, 10 and 100 µM are presented in Figure 2.

These data confirmed FLuc inhibition in vitro for seven of the 11 isoflavonoids. At the lowest isoflavonoid concentration tested of 1 μM, biochanin A and formononetin significantly inhibited FLuc (62.4%, 32.4%, respectively). For the intermediate isoflavonoid concentration of 10 μM, significant inhibition of FLuc was seen for glycitein, calycosin, and prunetin (30.1%, 66.2%, 40.7%, respectively). Finally, daidzein and genistein showed significant inhibition of FLuc at 100 µM (60.9%, 71.2%, respectively). Genistin, daidzin, glycitin, and S-equol were inactive against FLuc.

The IC_50_ values for FLuc inhibition were determined for all of the active isoflavonoids, along with the positive control, resveratrol, as given in Table 2 and shown in Appendix A. The most active compound was biochanin A, with an IC_50_ of 640 nM, while the IC_50_ for the resveratrol control was almost eight-fold higher at 4.94 µM. Similar FLuc inhibitory activities to resveratrol were seen for formononetin (3.88 µM) and calycosin (4.96 µM). The other active isoflavonoids had IC_50_ values >10 µM.

None of these isoflavonoids inhibited RLuc in vitro, as seen in Figure 2b. The positive control for RLuc inhibition, H-89, had an IC_50_ of 338.4 µM, with its dose–response curve shown in Appendix A.

### 2.3. Molecular Docking Calculations

To further rationalize these in vitro data, molecular docking of the isoflavonoids was carried out using the available crystal structures of FLuc and RLuc. Initially, the electrostatic properties of the FLuc binding site were analyzed. The pocket for the binding of ATP and the D-luciferin substrate is bent and is relatively narrow around the D-luciferin sub-pocket. It is also relatively hydrophilic and includes several hydrophobic amino acids (Appendix A). As expected, the contours of the calculated molecular interaction field using a hydrophobic probe reflected these characteristics, with several such regions indicated within the binding site. For example, two hydrophobic regions are shown in Figure 3c: one located around the amino acids Gly341 and Ile351; and the second encompassed by the amino acids His245, Ala348, Ala313, and Phe247. Further analysis by hydrogen bond donor and hydrogen bond acceptor probes pinpointed areas that were favorable for ligand–protein hydrogen-bond formation. A region of FLuc centered around the amino acid Thr343 was identified as particularly favorable (Figure 3a,b).

In docking experiments, the isoflavonoids showed highly comparable binding modes, and interacted exclusively with the FLuc D-luciferin binding pocket, with no indications of docking to the neighboring AMP sub-pocket. Figure 3d shows a docking pose for the most potent of these isoflavonoid inhibitors, biochanin A, as an example here. Amino acid Thr343 served as the anchoring residue, with the formation of a hydrogen bond with the hydroxylic substituent located on the core molecular scaffold. In addition, amino acids Phe247, Thr251, Ala348, and Ile351 established hydrophobic interactions with the phenolic moiety in all of these isoflavonoids. The binding poses of the additional compounds of the series are shown in the Appendix A for comparison (Appendix A). The influence of any particular FLuc crystal structure on the isoflavonoid docking pose outcomes was low, and all of the structures showed fully comparable binding poses for the D-luciferin sub-pocket (Figure 3e; Appendix A).

The initial docking experiments used a water-free FLuc binding site, and two orientations of the isoflavonoids were detected, with the main difference being the 180° rotation of the core bicyclic scaffold (Appendix A). To further investigate the binding properties of these isoflavonoids, another set of docking experiments was carried out, this time including water molecules W615 (PDB:3IES) and W709 (PDB:4E5D), which were indicated as important in a previous study [12]. Using these settings, only one of the orientations remained for the isoflavonoids. In this orientation, a carbonyl group on the isoflavonoid scaffold and Ser347 formed a hydrogen bond, which can be considered as another important interaction for successful molecular recognition of the ligand. The GoldScore values also favored this orientation, which was indicated as a more stable bound orientation (Figure 3d).

In contrast, RLuc operates via a completely different reaction mechanism, which is also reflected by the different binding site characteristics, as it is larger and more elongated than for FLuc. The RLuc binding site consists largely of hydrophobic amino acids, e.g., Trp156, Phe180, Phe261, and Phe262 (Appendix A). This is reflected in the extensive hydrophobic area that was detected by the H-probe molecular interaction field (Appendix A).

When the isoflavonoids were docked into the RLuc binding site, the poses generated showed no uniformity and were scattered across many possible orientations. On the other hand, a very uniform binding pose was obtained for the ensemble of the docking solutions for FLuc (Figure 4). Even the addition of water molecule W655 at the bottom of the RLuc binding site did not lead to any significant changes in the docking solutions. In addition, the GoldScore values were substantially lower for the binding modes of these isoflavonoids in FLuc vs. RLuc, which was in agreement with the results of the in vitro experiments performed.

## 3. Discussion

When studying the influence of specific compounds on gene expression using reporter-gene assays, it is essential to evaluate beforehand any interference of these compounds in the assay system. In the present study, we tested 11 isoflavonoids for inhibition of FLuc and RLuc using different approaches, including QSAR modeling, cell-lysate in vitro luciferase inhibition assays, and in silico molecular docking.

The InterPred model for the prediction of FLuc interference is a newly developed QSAR model that builds on the data for in vitro FLuc inhibition assays from the Tox21 Consortium library. Although it was cross-validated and showed near-perfect performance on the training set (Matthews correlation coefficient, 0.996 ± 0.002), some discrepancies were observed between its predictions here and the in vitro results for FLuc inhibition by these isoflavonoids. The lowest likelihood of interference (i.e., InterPred, green class) was not predicted for any of these 11 isoflavonoids, although four were inactive at concentrations as high as 100 µM in the in vitro assays. On the other hand, the isoflavonoids that were predicted to have the highest likelihood of interference (i.e., daidzein, genistein, glycitein, prunetin, biochanin A, calycosin, and formononetin) were also seen as FLuc inhibitors in vitro. These isoflavonoids share structure similarities, and therefore the in silico predictions of interference were similar too. However, the IC_50_ values determined in vitro indicated different activities for these isoflavonoids within the InterPred highest predicted FLuc inhibition class (i.e., red). The highest in silico probabilities of interference were for biochanin A, formononetin, and genistein, although genistein showed much weaker inhibitory activity in vitro compared to biochanin A and formononetin. Conversely, calycosin showed one of the lowest in silico likelihoods for this class (i.e., InterPred, red), while it was a more potent inhibitor in vitro than some of the compounds with higher in silico likelihoods in this class. However, overall, InterPred performed well, as all of the isoflavonoids that were predicted to be in the highest likelihood of interference class indeed showed relevant inhibitory activity in vitro.

As the purchase of the purified reporter protein for in vitro inhibition assay is relatively costly, it is necessary to develop cheaper alternative methods for the testing for inhibition in such reporter-gene assays. To some extent, this was addressed in the OECD test guidelines for such compounds. Thus, TG 455 suggests a protocol to test for false-positive results for human estrogen receptor-α agonism by blocking transcription of the reporter gene (*FLuc*) with 4-hydroxytamoxifen [19]. In this way, it can be clarified whether the luminescence upon exposure to a specific compound is mediated through human estrogen receptor-α or not.

However, blocking a signaling pathway at a transcriptional level with a known antagonist does not help to define interference in these reporter-gene assays due to inhibition of the protein expressed by the reporter gene (i.e., luciferase). Here, we achieved this in a rapid, technically simple, and cost-effective way using the AR-EcoScreen cell lysates. This cell line is included in OECD TG 458 and expresses both FLuc and RLuc [10]. By using the same culture conditions and reagents, we believe that the method we describe here provides more relevant luciferase inhibition information than assays on isolated enzymes. Indeed, the IC_50_ values of the positive controls were higher in the present cell-lysate assay system than reported previously. Resveratrol had an IC_50_ of 4.94 µM in this cell-lysate assay, while an IC_50_ of 2 µM was reported for isolated FLuc [14]. The same previous study also showed that the IC_50_ for resveratrol was independent of FLuc concentration [14], which can indeed vary among cell lysates in biological repeats. Hence, this difference in IC_50_ might be attributable to the detection reagents used for the commercially available luciferase, which are often designed to prevent inhibition and contain an excess of a substrate and coenzyme A, to mitigate the formation of multisubstrate adduct inhibitors [3]. The dependence of RLuc inhibition on substrate concentration was also shown in a study that reported an IC_50_ of 10 µM for the positive control of H-89 in the presence of 10 µM coelenterazine, whereas H-89 had an IC_50_ of 338.4 µM for RLuc in the present study [24]. This unexpectedly high IC_50_ of the positive control for RLuc inhibition indicates that the experimental combination used for the reporter cell lysate and the commercially available dual-luciferase detection reagent is less susceptible to known RLuc inhibitors than to known FLuc inhibitors.

None of the isoflavonoids studied here with this cell-lysate in vitro assay showed inhibition of RLuc. The FLuc inhibition of these isoflavonoids is evidently dictated by their structural properties, where some of the structural modifications significantly increased the inhibition of FLuc, such as O-methylation of the basic isoflavonoid structure on the A-ring (i.e., glycitein, prunetin), and more prominently, on the B-ring (i.e., biochanin A, calycosin, formononetin). These isoflavonoids have versatile biological activities, which have often been studied in reporter-gene assays with FLuc [25,26,27,28]. Caution is therefore needed when interpreting such results, as FLuc inhibitors can give false-positive results, as well as false-negative results (i.e., where they attenuate the luminescence signal when their transcription is actually activated). The nonmethylated isoflavonoids (i.e., daidzein, genistein) showed weaker inhibitory activities than the O-methylated isoflavonoids, while the glucosides (i.e., daidzin, genistin, glycitin) and the representative of the isoflavonoids (S-equol) did not show inhibition of FLuc at all.

These results from the cell-lysate in vitro assays were further supported by in silico molecular docking in the active sites of FLuc and RLuc. Taken together, the in silico docking analysis suggested that these isoflavonoids inhibit FLuc activity by interacting with its D-luciferin binding (sub-)pocket. The very uniform docking modes for these isoflavonoids suggested that their successful molecular recognition is defined by two hydrogen bonds between the ligand and the FLuc residues Thr343 and Ser347, along with several hydrophobic interactions. The proposed binding model also further rationalizes the lack of inhibitory activity of the remaining four isoflavonoids. Here, the inactive glucosides daidzin, genistin, and glycitin, contain a bulky glucose molecule that might hinder the formation of the hydrogen bond with Thr434. Furthermore, the absence of a carbonyl group in S-equol leads to the loss of the hydrogen bond with Ser347.

There was also little influence of the luciferase crystal structures used on the obtained isoflavonoid docking poses. The three-dimensional structural alignments of the FLuc X-ray structures demonstrated that the luciferase binding sites remain relatively unperturbed after binding of chemically diverse native ligands. This, in turn, provides more confidence for the assumption that substantial rearrangement within the binding site is not likely to occur also for the newly discovered inhibitors. As these isoflavonoids docked exclusively into the D-luciferin binding pocket, we speculate that their inhibition mechanism is through competition with the native D-luciferin substrate. This appears to be in line with the previously reported increased responses of selected isoflavonoids [29], which can be attributed to the well-known inhibitor-based stabilization of FLuc and the subsequent competition with the excess substrate in the commercial reagents.

Finally, this molecular docking also provided plausible structure-based rationalizations for the observed selectivity of the studied isoflavonoids towards FLuc. Namely, the narrower D-luciferin binding pocket can more easily adopt a predominantly planar structure similar to these isoflavonoids, which also share some level of structural similarity to the native D-luciferin molecule. On the other hand, the broader RLuc binding site appears to be less favorable for the uniform and thermodynamically favorable bound ligand conformation.

## 4. Materials and Methods

### 4.1. In-Silico Prediction with InterPred

InterPred webserver (version 1.0, accessed on 3 August 2020, National Institute of Environmental Health Sciences, Durham, NC, USA) is a platform for the prediction of interference in cellular and biochemical assays that use fluorescence or luminescence as the detection technology [23]. InterPred includes 17 QSAR models that were developed using machine learning, and are based on data from more than 8,000 unique structures in assays contained in the Tox21 Consortium library, i.e., luciferase inhibition assays (tox21-luc-biochem-p1), and autofluorescence assays in the HepG2 and HEK-293 cell lines, and the cell-free assay formats (tox21-spec-hepg2-p1, tox21-spec-hek293-p1, respectively) [1,23]. The intent of InterPred is to enable the identification of false-positive results due to autofluorescence and inhibition of luciferase in large high-throughput screening assays.

The selected isoflavonoids were submitted to the InterPred model for luciferase interference in the SMILES format, as obtained from PubChem, and the structures were prepared for QSAR modeling by removal of hydrogens, salt fragments, metal disconnection, sanitization, desolvation, and stereochemistry processing, and computing of the one-dimensional, two-dimensional, and physicochemical descriptors [23].

The results were returned in the form of likelihoods that were in the range of 0 to 1, with the associated standard deviations. Values closer to 1 define higher likelihoods that a compound will cause FLuc interference. The results from InterPred are also color-coded based on the numerical values of the likelihood of FLuc inhibition, as follows: green, likelihood < 0.25 (i.e., lowest likelihood); yellow, 0.25 ≤ likelihood < 0.50; orange, 0.5 ≤ likelihood < 0.75 (i.e., moderate probabilities of inhibition); and red, likelihood ≥ 0.75 (i.e., the highest likelihood of inhibition).

### 4.2. Chemicals

Glycitein (≥95%; CAS 40957-83-3), prunetin (≥95%; CAS 552-59-0), biochanin A (≥99%; CAS 491-80-5), formononetin (≥99%; CAS 485-72-3), and glycitin (≥95%; CAS 40246-10-4) were from Extrasynthese (Lyon, France). Daidzien (≥98%; CAS 486-66-8), genistein (≥98%, CAS 446-72-0), daidzin (≥95%; CAS 552-66-9), and genistin (≥95%; CAS 529-59-9) were from Sigma-Aldrich (St. Louis, MO, USA). Calycosin (99%; CAS 25389-94-0) was from Phytolab (Vestenbergsgreuth, Germany), and S-equol (97%; CAS 531-95-3) was from Toronto Research Chemicals (Toronto, Canada). The FLuc positive control was resveratrol (>99%; CAS 501-36-0; Sigma-Aldrich, St. Louis, MO, USA), and the RLuc positive control was H-89 (99.34%; CAS 130964-39-5; MedChemExpress, Monmouth Junction, NJ, USA). The isoflavonoids were dissolved in cell-grade dimethylsulfoxide (DMSO; 99.9%; CAS 67-68-5; Sigma-Aldrich, St. Louis, MO, USA) and serially diluted to prepare 100× stock solutions. All of the isoflavonoids were screened in the in vitro FLuc and RLuc inhibition assays at 1, 10, and 100 µM final concentrations. Dose–response curves were generated for the active isoflavonoids and for the positive controls to determine the IC_50_ values.

### 4.3. Cell Culture and Preparation of AR-EcoScreen Cell Lysate

The AR-EcoScreen cell line was derived from a Chinese hamster ovary cell line that was stably transfected with the androgen receptor and the FLuc and Rluc reporter genes. These AR-EcoScreen cells are used for detection of androgen receptor agonist and antagonist activities (OECD TG 458), where the expression of FLuc is androgen-receptor-driven, while Rluc is expressed constitutively, to allow detection of cytotoxicity [10]. The AR-EcoScreen cell line was purchased from the Japanese Collection of Research Bioresources (JCRB) Cell Bank (JCRB1328, Osaka, Japan), and was maintained in phenol-red-free Dulbecco’s modified Eagle’s medium/F-12 (Gibco, Waltham, MA, USA), supplemented with 5% dextran-charcoal-stripped fetal bovine serum (Gibco, Waltham, MA, USA), 200 μg/mL zeocin (Invivogen, Toulouse, France), 100 μg/mL hygromycin B, 100 U/mL penicillin, and 100 μg/mL streptomycin (all from Sigma-Aldrich, St. Louis, MO, USA).

Preparation of cell lysates for use in the luciferase inhibition assays was through seeding of 4–7 ×10^6^ cells into a 150 cm^2^ growth flask (an equivalent surface density to that indicated in OECD TG 458 [10]) for 24 h at 37 °C under 5% CO_2_. Upon incubation, dihydrotestosterone (Sigma-Aldrich, St. Louis, MO, USA) diluted in DMSO was added at 10 nM final concentration. This was followed by another 24-h incubation, removal of medium, and addition of 9.4 mL reporter lysis buffer (Promega, Madison, WI, USA). The growth flask was then stored at −80 °C until use.

### 4.4. Cell-Lysate Luciferase Inhibition Assay

The cell lysate was thawed at room temperature. Stock solutions of the isoflavonoids and positive controls were dissolved in DMSO and 10-fold serially diluted in the cell lysate to the final concentration with 1% DMSO. Twenty microliters of each isoflavonoid in the cell lysate were transferred to 96-well plates in triplicate and incubated for 30 min at room temperature. Then, 30 µL Luciferase Assay Reagent II (contained in the Dual-Luciferase Reporter Assay System; Promega, Madison, WI, USA) was added to each well, and the FLuc luminescence was measured using a microplate reader (Synergy 4 hybrid; BioTek, Winooski, VT, USA). The addition of 30 µL of Stop & Glo Reagent (contained in the Dual-Luciferase Reporter Assay System) followed, and the RLuc luminescence was recorded using the microplate reader (Synergy 4 hybrid; BioTek, Winooski, VT, USA).

### 4.5. Data Analysis

The data from the in vitro luciferase inhibition assays in the AR-EcoScreen cell lysate were processed by first subtracting the background value (i.e., reporter lysis buffer alone) and then normalization to the vehicle control (1% DMSO), and are expressed as means ± standard deviation of three independent repeats carried out as technical triplicates. GraphPad Prism 8.0 (GraphPad Software, San Diego, CA, USA) was used for the calculation of the inhibitory concentrations at 50% activity (IC_50_), and for the relevant Figures and statistical analysis. Statistical significance as compared to the vehicle control, considered: *, *p* ≤ 0.05; **, *p* ≤ 0.01; and ****, *p* ≤ 0.0001, using one-way ANOVA, followed by post-hoc Dunnett’s tests.

### 4.6. Molecular Docking Calculations

Molecular docking of the isoflavonoids in the available X-ray structures of FLuc containing ligands PTC124-AMP (PDB: 3IES) [15], aspulvinone J-CR (PDB: 3RIX) [30], and benzothiazole (PDB: 4E5D) [12] was performed using the GOLD docking program [31]. Protein preparation, structural alignments, and comparisons of all three PDB structures were carried out within the Hermes environment. In the structural alignments, 3IES was used as the reference protein. Hydrogen atoms were added to all three proteins using the default settings. The PDB structures were initially stripped of all of the water molecules; however, additionally docking calculations were performed that included water molecules W615 for PDB:3IES and W709 for PDB:4E5D. These water molecules are considered to have important roles in the ligand–protein recognition [12]. The three-dimensional conformations of the isoflavonoids were generated in ChemBio3D Ultra interface and were geometrically optimized prior to docking calculations by applying the MMFF94 force field. The binding site for FLuc was defined within a radius of 6 Å around the coordinates of the bound ligands. Each molecule was docked into the binding site 10 times, with the following settings of the genetic search algorithm: population size, 100; selection pressure, 1.1; number of operations, 100,000; number of islands, 5; niche size, 2; crossover frequency, 95; mutation frequency, 95; and migration frequency, 10. The spins of the waters were allowed to change during the docking calculations. The GoldScore scoring function was used to define the favorability of the generated binding poses. The GOLD docking tool [32] settings were validated by successfully redocking all three of the X-ray ligands (PTC124-AMP, aspulvinone J-CR, benzothiazole) into the FLuc binding site. This provided more confidence in the docking parameters used (Appendix A).

The molecular docking into the RLuc crystal structure used the coelenteramide ligand (PDB:2PSJ) [33]. The protein was prepared as described above. The active site comprised a 6 Å radius around the reference ligand. As well as docking into the water-free binding site, the calculations were also performed with the inclusion of water molecule W655, with the same genetic search algorithm settings and GoldScore scoring function as described for FLuc. Validation redocking of coelenteramide to RLuc was successful, as the calculated ligand binding pose was close to the experimentally determined ligand orientation (Appendix A).

The results of the docking calculations are visualized in LigandScout [34]. This program was also used for the Apo site analysis using the default settings. During this task, various molecular probes (i.e., hydrogen bond acceptor, hydrogen bond donor, positive ionizable, negative ionizable, hydrophobic probes) scanned the RLuc and FLuc binding sites to generate contours of the corresponding molecular interaction fields (MIFs).

## 5. Conclusions

Using a combination of in silico and in vitro approaches, we demonstrated that RLuc is an appropriate substitute for FLuc in the testing of isoflavonoids, as FLuc was very susceptible to inhibition by the isoflavonoids selected here. On the other hand, none of these isoflavonoids inhibited Rluc in the cell-lysate in vitro luciferase inhibition assay. Furthermore, this study outlines the need for the use of prior testing of reporter cell lysates in routine screening for luciferase inhibition by compounds under investigation, such as those described here (i.e., these isoflavonoids), to exclude any interference in the assay system by such compounds in the reporter-gene assays.

## Figures and Tables

**Figure 1 ijms-22-06927-f001:**
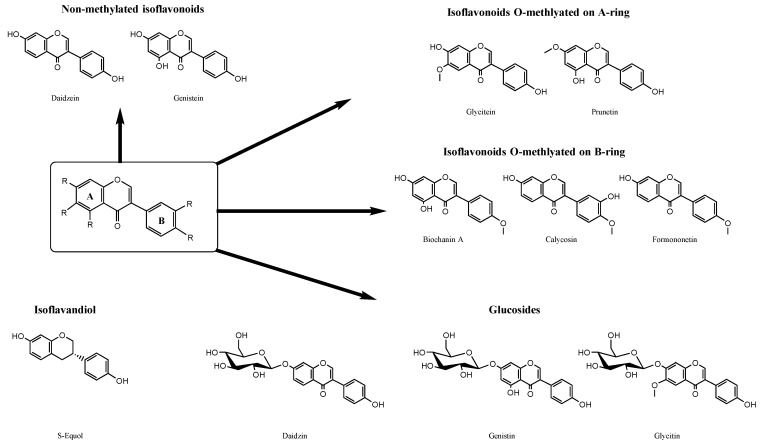
Modifications to the basic isoflavonoid structure for the selected isoflavonoids.

**Figure 2 ijms-22-06927-f002:**
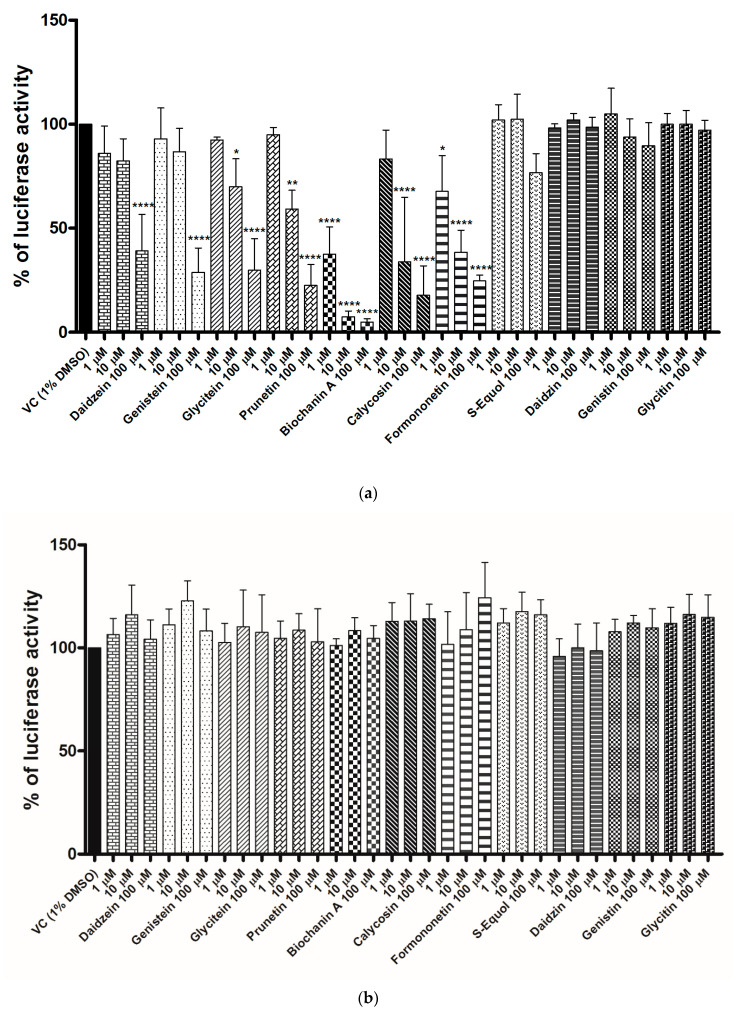
In vitro screening for inhibition of luciferase activities by the isoflavonoids. (**a**) Firefly luciferase activity. (**b**) *Renilla* luciferase activity. Data are defined as percentages of the vehicle control (VC) of 1% dimethylsulfoxide in the AR-EcoScreen cell lysates upon inclusion of the isoflavonoids (1, 10, 100 μM). Data are means ± standard deviation of three independent repeats in triplicates. *, *p* ≤ 0.05; **, *p* ≤ 0.01; ****, *p* ≤ 0.0001; vs. VC (one-way ANOVA, post-hoc Dunnett’s tests).

**Figure 3 ijms-22-06927-f003:**
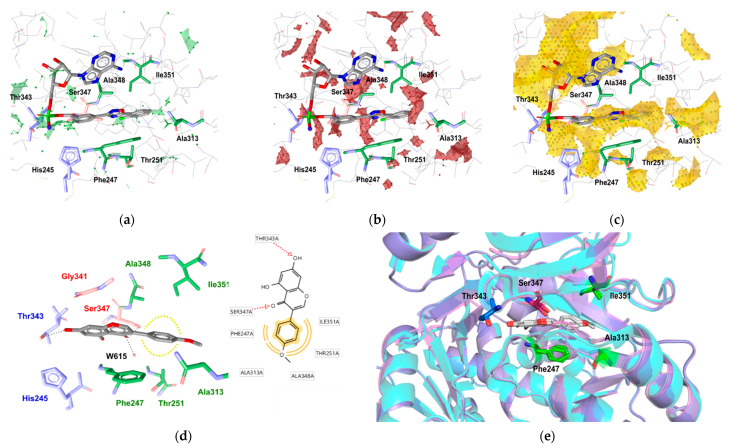
(**a**–**c**) Calculated molecular interaction fields using hydrogen bond donor (**a**; green), hydrogen bond acceptor (**b**; red), and hydrophobic (**c**; yellow) probes in the firefly luciferase binding site with the bound PTC124-AMP ligand shown (PBD:3IES). (**d**) Representative binding pose of biochanin A (left) and the two-dimensional scheme of the observed intermolecular interactions (right). (**e**) Three-dimensional alignment of the firefly luciferase crystal structures 3IES, 3RIX, and 4E5D, with representative predicted binding modes of biochanin A in each structure.

**Figure 4 ijms-22-06927-f004:**
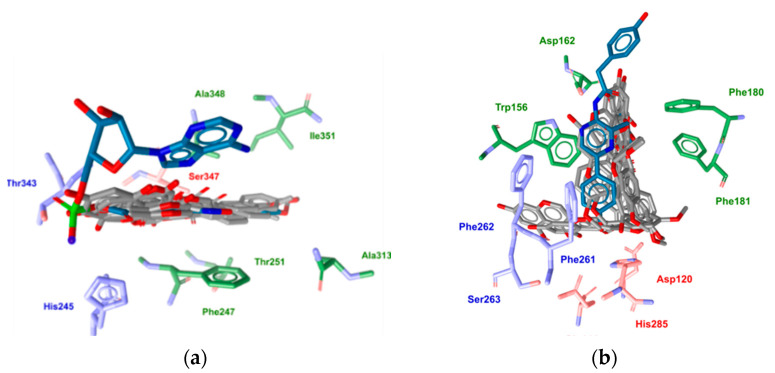
Comparisons of the ensembles of docking solutions obtained for biochanin A for the firefly luciferase (PDB: 3IES) (**a**) and *Renilla* luciferase (PDB: 2PSJ) (**b**) binding sites.

**Table 1 ijms-22-06927-t001:** Results of the InterPred predicted interference with firefly luciferase for the positive control (resveratrol) and the 11 selected isoflavonoids, according to their structural grouping (Figure 1). The predicted likelihood of interference is expressed from 0 to 1 (closer to 1, higher likelihood of interference), with color-coding also shown (see Methods).

Structural Group	Selected	Prediction	
	Isoflavonoid	Likelihood of Interference	Color Class
Stilbene	Resveratrol ^1^	0.92 ± 0.07	Red
Non-methylated	Daidzein	0.87 ± 0.11	Red
	Genistein	0.93 ± 0.04	Red
O-methylation (A)	Glycitein	0.87 ± 0.05	Red
	Prunetin	0.88 ± 0.05	Red
O-methylation (B)	Biochanin A	0.94 ± 0.04	Red
	Calycosin	0.86 ± 0.07	Red
	Formononetin	0.94 ± 0.04	Red
Isoflavandiol	S-Equol	0.73 ± 0.04	Orange
Glucosides	Daidzin	0.40 ± 0.10	Yellow
	Genistin	0.40 ± 0.10	Yellow
	Glycitin	0.34 ± 0.11	Yellow

^1^, positive control.

**Table 2 ijms-22-06927-t002:** In vitro determined IC_50_ values for firefly luciferase inhibition for the positive control (resveratrol) and the 11 selected isoflavonoids, according to their structural grouping (Figure 1).

Structural Group	Selected Isoflavonoid	IC_50_ (µM)
Stilbene	Resveratrol	4.94
Non-methylated	Daidzein	51.44
	Genistein	36.89
O-methylation (A)	Glycitein	26.40
	Prunetin	16.27
O-methylation (B)	Biochanin A	0.64
	Calycosin	4.96
	Formononetin	3.88

## Data Availability

Molecular docking was performed using the program Gold available within Hermes 1.7.0 environment (Cambridge Crystallographic Data Centre (CCDC), Cambridge, UK). Results were further visualized in LigandScout, version 4.4.3 (Inte:Ligand, Vienna, Austria). The starting structures were obtained from the Protein Data Bank (PDB) public structure database. All procedures and workflows are described in the Methods section. Further data supporting the findings of this study are available from the corresponding author upon reasonable request.

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
