# Peer review of "Evaluation of Firefly and Renilla Luciferase Inhibition in Reporter-Gene Assays: A Case of Isoflavonoids"

_ijms, 2021, doi:10.3390/ijms22136927_

Round 1

Reviewer 1 Report

The authors present a very interesting manuscript on putative inhibition of isoflavonoids on two different types of luciferase. The results show that one of the two luciferases is not inhibited by these compounds and is suggested for use in reporter gene assays testing isoflavonoids. Also the structural and functional background of inhibition has been investigated. 

The authors present a very well written manuscript showing highly interesting data which are well presented. 

The manuscript is ready for publication in its present form. The authors should check one methodical aspect. Is it really correct that the growth flasks were stored at -80°C?

Author Response

The authors are grateful for the positive opinion of the work they have done and described in this article.

The manuscript was re-read carefully due to possible typos and grammatical errors.

Reviewer question:

“The manuscript is ready for publication in its present form. The authors should check one methodical aspect. Is it really correct that the growth flasks were stored at -80°C?”

Authors' response:

The storage of the prepared cell lysate is correctly described. The lysis reagent added to the growth flask is a mild lysis agent and requires a single freeze-thaw cycle at -80°C to achieve complete cell lysis as per manufacturer’s instructions. Hence, the growth flask had to be frozen before the lysate could be transferred to a different container. To ensure the stability of enzymes in the lysate, the growth flask was not thawed until needed for the inhibition assay. However, the lysate was not stored at -80°C for longer periods of time, as it was used within a week.

Reviewer 2 Report

In this work, by Maša Kenda and colleagues, authors addressed an important issue related to reporter gene assays, based of firefly/renilla luciferases, used in screening campaigns to identify drug-based inhibitors, which are the source of potential false-positive results.

They have evaluated isoflavonoids (as an example) for inhibition of firefly luciferase using different approaches, including QSAR modeling, cell-lysate in-vitro luciferase inhibition assays, and in-silico molecular docking. Particularly, the in vitro assay revealed how renilla luciferase is not prone to inhibition by isoflavonoids compared to firely luciferase, further stressing the importance of choosing the right reporter gene in a specific screening campaign. Despite some discrepancy between the QSAR modeling (InterPred) and the in vitro data, molecular docking calculations indicated that isoflavonoids interact favorably with the D-luciferin binding pocket of firefly luciferase, but not with with the active site of RLuc, confirming the in vitro-obtained results.

The manuscript is well written and the data presented reasonable, therefore I recommend it to be accepted in the present form.

Author Response

The authors are grateful for the positive opinion of the work they have done and described in this article.